# Analysis of the Relationship between Socioeconomic Status and Incidence of Hysterectomy Using Data of the Korean Genome and Epidemiology Study (KoGES)

**DOI:** 10.3390/healthcare10060997

**Published:** 2022-05-27

**Authors:** Yung-Taek Ouh, Kyung-Jin Min, Sanghoon Lee, Jin-Hwa Hong, Jae Yun Song, Jae-Kwan Lee, Nak Woo Lee

**Affiliations:** 1Department of Obstetrics and Gynecology, School of Medicine, Kangwon National University, Chuncheon 24289, Korea; oytjjang@gmail.com; 2Department of Obstetrics and Gynecology, Korea University Ansan Hospital, Ansan 15355, Korea; nwlee@korea.ac.kr; 3Department of Obstetrics and Gynecology, Korea University Anam Hospital, Seoul 02841, Korea; mdleesh@gmail.com (S.L.); sjyuni105@gmail.com (J.Y.S.); 4Department of Obstetrics and Gynecology, Korea University Guro Hospital, Seoul 08308, Korea; jhhong93@korea.ac.kr (J.-H.H.); jklee38@gmail.com (J.-K.L.)

**Keywords:** hysterectomy, socioeconomic status, education, income

## Abstract

Hysterectomy remains a frequent gynecologic surgery, although its rates have been decreasing. The aim of this study was to investigate whether socioeconomic status affected the risk of hysterectomy in Korean women. This prospective cohort study used epidemiologic data from 2001 to 2016, from the Korean Genomic and Epidemiology Study (KoGES). Multivariate logistic regression analyses were performed to estimate the association between household income or education level and hysterectomy. Among 5272 Korean women aged 40–69 years, 720 who had a hysterectomy and 4552 controls were selected. Variable factors were adjusted using logistic regression analysis (adjusted model). Adjusted odds ratios (aORs) for insurance type and hysterectomy were not statistically significant. The aOR was 1.479 (95% confidence interval (CI): 1.018–2.146, *p* < 0.05) for women with education of high school or lower compared to college or higher. Women whose monthly household income was <KRW 4,000,000 had a higher risk of undergoing hysterectomy than women whose monthly household income was ≥KRW 4,000,000 (aOR: 2.193, 95% CI: 1.639–2.933, *p* < 0.001). Overall, the present study elucidated that lower socioeconomic status could increase the incidence of hysterectomy. Our results indicate that the implementation of stratified preventive strategies for uterine disease in those with low education and low income could be beneficial.

## 1. Introduction

Hysterectomy is one of the most frequent gynecological surgeries in more economically developed countries [1]. According to the 2012 Health Data from the Organization for Economic Cooperation and Development, 329.6 out of 100,000 women in Korea underwent hysterectomy in 2010, which was the highest among 34 developed countries [2]. According to a study conducted in Korea in the 2000s, the frequency of hysterectomy was then increasing [3]. Hysterectomy is known to be linked to poor quality of life and poor health outcomes [4,5]. About 20 to 45% of women who undergo hysterectomy are in their 50s and 60s [6]. They mostly had benign diseases, including uterine leiomyoma, abnormal uterine bleeding, endometriosis, and prolapse [3]. Although there are positive results for improving quality of life after hysterectomy due to vaginal bleeding or dysmenorrhea, it also has negative effects, so it should be carefully considered. It was reported that hysterectomy could cause postoperative psychiatric morbidity, such as depression and psychiatric referrals [7].

In Korea, 97% of the total population is enrolled in Korea National Health Insurance to receive medical care, and the government pays about 60% of the total cost. The remaining 3% is covered under the Medical Aid Program, which is fully supported by the government. It is well known that socioeconomic status (SES) is associated with various health outcomes [8]. Studies have been conducted on the relationship between socioeconomic factors and hysterectomy. About 80% of hysterectomy might be for non-oncologic diseases. Some studies have suggested that ethnic/racial differences are related to hysterectomy [9,10,11]. However, income and education level have not been studied in detail yet. In studies from other countries, hysterectomy was associated with lower SES or lower levels of education [12,13,14]. This subject has not been considered in Korea. To reduce excessive hysterectomy, it is necessary to identify the high-risk group for hysterectomy in advance and implement conservative treatment at an early stage.

Thus, the objective of this study was to investigate the association between SES (household income, education level, and insurance type) and hysterectomy using data from the Korean Genome and Epidemiology Study (KoGES). In this study, we implemented a large population data set by coordinating various influential factors.

## 2. Materials and Methods

### 2.1. Study Population

We used data from the Korean Genome Epidemiology Study (KoGES) prospective cohort study which was initiated in 2001 and supported by the National Genome Research Institute of Korea Centers for Disease Control and Prevention. The KoGES study investigated environmental and genetic factors affecting common chronic disease in Korean people [15]. Within the framework of the KoGES, two community-based cohort studies began in 2001. These cohorts have been followed up biennially since 2001. Subjects were followed up until the 7th regular survey conducted in 2015 and 2016. Participants aged 40–69 years included residents of urban (Ansan) and rural (Ansung) areas. This has been described in a previous study [16]. Of 5333 women, participants were excluded if they had missing information during study periods or previous hysterectomy or uterine malignancy before enrollment. Finally, a total of 5272 women was included in this analysis.

### 2.2. Data Collection

Information regarding sociodemographic status and lifestyle, anthropometric measurements, and medical history of participants was obtained by trained interviewers using questionnaire [15]. Questionnaire surveys, clinical investigations, and physical examinations were performed during baseline and follow-up assessments. History of obstetrics and gynecology such as parity, age of menarche, first pregnancy, first delivery, and breast feeding were also investigated. Postmenopausal hormone replacement therapy was also investigated. The data did not contain information about underling gynecological diseases including uterine fibroids, adenomyosis or ovarian diseases. Obstetric complications such as placenta previa or uterine atony were also not collected.

Anthropometric measurements were obtained by trained healthcare providers. Height and body weight were measured for participants wearing light clothing and barefoot. Body mass index (BMI) was calculated as weight (kg) divided by the square of the height (m^2^).

### 2.3. Statistical Analysis

SES of participants was defined according to their household income, education level, and insurance type. Characteristics of enrolled participants were categorized as follows: education level (elementary, middle, high school, and college or higher), marital status (no, married, widow, separated, and divorced), household income ((in KRW) <50, 50–100, 100–150, 150–200, 200–300, 300–400, 400–600, ≥600 × 10,000), insurance (national insurance, medicaid).

Data are expressed as mean ± standard deviation for continuous variables or frequency and percentage for categorical variables. Logistic regression analysis was assessed to estimate the odds ratios (ORs) and 95% confidence intervals (CIs) of the risk of hysterectomy. Chi-square tests were used to compare education level, marital status, household income, insurance, breastfeeding, and postmenopausal hormone replacement therapy. Student’s *t*-tests were used to compare age, parity, age of menarche, age of first pregnancy/delivery, and BMI. All statistical analyses were performed using SAS 9.2 software (SAS Institute Inc., Cary, NC, USA). A two-sided *p*-value of less than 0.05 was considered statistically significant.

## 3. Results

In total, 720 women with hysterectomy and 4552 women without hysterectomy were identified (Table 1). Compared to those without hysterectomy, those with hysterectomy were more likely to be younger (*p* < 0.001). The mean age was 54.63 ± 8.90 years old for women with hysterectomy and 61.23 ± 9.93 years old for women without hysterectomy. The risk of hysterectomy was higher among married women than among widows (*p* < 0.001). Women with hysterectomy had higher household income (*p* < 0.001) (Figure 1). The risk of hysterectomy was lower in women with a household income ≥ KRW 4,000,000 per month than in women with household income less than KRW 1,000,000 per month. Women with hysterectomy were more likely to be younger at first delivery (*p* < 0.001), have higher BMI (*p* = 0.001), and have more postmenopausal hormone replacement therapy (*p* < 0.001).

Table 2 shows the association of SES status with the risk of hysterectomy. Significant differences were found in education level and household income between hysterectomy and non-hysterectomy groups. The proportion of participants with hysterectomy was increased in women with education level lower than high school compared to that in women with college education or higher (aOR: 1.479, 95% CI: 1.018–2.146, *p* < 0.05), after adjusting for covariates, including age, BMI, household income, insurance, and education level. Lower household income was also associated with increased risk of undergoing hysterectomy (aOR: 2.193, 95% CI: 1.639–2.933, monthly income ≥ KRW 4,000,000 vs. monthly income < KRW 4,000,000, *p* < 0.001). However, insurance type was not associated with hysterectomy (*p* = 0.090).

## 4. Discussion

In the present study, low SES was associated with a higher risk of hysterectomy. Hysterectomy was more likely to be prevalent in women with lower education level (high school or lower) and lower household income (<KRW 4,000,000 per month). These results were derived from a large, prospective community-based general population. Furthermore, to the best of our knowledge, our study is the first one to investigate the effect of SES on hysterectomy in Korea. 

SES has been suggested to contribute to health status through the intake of nutrients, which could increase hysterectomy in the lower SES group by increasing the risk of uterine leiomyoma. Previous meta-analyses have shown that high SES is associated with high intake of dietary fiber, vitamin C, folate, beta-carotene, calcium, and iron [17]. Women with lower SES tend to have fewer fruits and vegetables with a high-fat and low-cost diet [18,19]. Many nutrients and dietary habits are associated with increased risk of uterine leiomyoma [20]. A low intake of fruit and green vegetables and pollutants ingested with food are associated with the development of uterine leiomyoma [21,22]. Two studies performed on Chinese women found that the intake of vegetables and fruits can decrease the risk of uterine fibroids through the protective effects of dietary phytochemicals, which could regulate extracellular matrix deposition, cell proliferation, and angiogenesis [23,24]. In addition, inverse correlation between serum vitamin D level and risk of myoma has been confirmed in previous studies [25,26]. It is unclear how exactly dietary products and nutrients affect the progression of uterine fibroids.

A previous study found that the risk of hysterectomy was higher for women with a high-school education or lower than women with a high-school education or higher [13]. A high level of education provides an opportunity to obtain a lot of health-related information, which, in turn, leads to health-promoting lifestyle changes. A low level of education not only affects the level of general health, health understanding, and ability to navigate medical systems, but also affects decision-making ability, which, in turn, could affect the quality of treatment received through socioeconomic disadvantages [27]. Women with lower levels of education are more likely to work in low-income jobs that could negatively affect their health [28]. Occupations involving working in shifts or doing piecework might have an impact on abnormal uterine bleeding [29]. Additionally, they are at an increased risk of becoming overweight and obese, which can lead to a higher risk of chronic metabolic diseases in general, as well as hysterectomy [30]. 

We found that hysterectomy was more prevalent in the lower-income group. One possible explanation might be because women with a lower income would seek medical care later in the disease progression or may not be candidates for conservative management. Various alternative treatments for hysterectomy have emerged in Korea. For benign uterine disease, hysterectomy can be prevented through less aggressive strategies, such as conservative management, hormone therapy, and levonorgestrel intra-uterine devices through regular screening, which is more accessible to women with higher income. Furthermore, laparoscopic myomectomy is an alternative treatment for fibroids in women who want to preserve the uterus. In addition, the perception of hysterectomy has recently changed [31]. Nevertheless, women with lower incomes prefer hysterectomy because it is cheaper and safer without the risk of recurrence. In addition, the demand for alternative therapies, such as uterine artery embolization or high-intensity focused ultrasound (HIFU), has been increasing recently [32]. Korean women with low income prefer hysterectomy to alternatives because insurance does not cover alternative treatments.

Although Korean women with low income prefer hysterectomy in this study, insurance type was not associated with hysterectomy. Women with Medicaid generally have lower incomes, but the burden is light because the fees of treatment and surgery paid by them is very low. However, there are also many physically or mentally disabled women who have difficulty in visiting hospitals [33]. In fact, in Korea, a social problem has been considered about the low accessibility to healthcare among the Medicaid population [34]. Further studies on these issues should be conducted in the future.

Early menarche increases the risk of developing uterine fibroids and endometriosis [35,36,37], which are common indications for hysterectomy. Although the exact mechanism has not been elucidated, it might be because the frequency of stimulation by hormones is higher due to a greater exposure to the menstrual cycle [38]. Early menarche may also represent early sexual activity, which may affect marriage or early first birth [39]. In this study, there was no relationship between age at menarche or age at first birth and hysterectomy. Thus, further research is needed.

The present study had several limitations. First, because of the limitation of a cross-sectional study, it was difficult to prove a clear causal relationship between SES and hysterectomy. Second, it was impossible to know the causal disease of hysterectomy because we did not have patients’ medical records or diagnosis code at the time of surgery. For example, in addition to uterine myoma or adenomyosis, hysterectomy could be performed for malignant diseases, such as cervical cancer and endometrial cancer. However, it was not possible to obtain such information. Third, these data were collected using an individual questionnaire survey. Hence, the precision of the data used in this study might not be high. Additional studies involving a larger number of participants are required to validate the relationship between low SES and the prevalence of hysterectomy in the general population. Nevertheless, this study provides valuable information about the relationship between SES and hysterectomy. This study also has some strengths. First, it was performed using a large sample of the Korean population. Second, age and BMI as factors were adjusted to investigate the independent association between SES and hysterectomy.

Although our study found that the risk of hysterectomy is increased in women with lower education or lower incomes, the mediating effect of SES on hysterectomy cannot be explained. The probable possibilities are: (1) lower screening rates, (2) increased prevalence of uterine diseases, including uterine fibroids, and (3) lack of medical knowledge. Further research on this is needed, and policy suggestions can be considered. First, although Korea is implementing national screening for all people, a strengthened policy is needed to carefully implement screening programs for the socially vulnerable. Second, education and training on medical knowledge should be strengthened for all people. Finally, consensus on the establishment of domestic guidelines for hysterectomy indications is needed to prevent indiscriminate hysterectomy.

Several studies found a significant relationship between socioeconomic factors and health problems in women [8,33,40]. Significant disparities in healthcare services in women with endometriosis according to SES were found [41]. The risk of infant death was significantly higher among infants of parents with the lowest level of education [42]. In addition, the risk of death was higher among infants with no maternal education level, but paternal education level [42]. Further studies may be needed to include other socioeconomic factors, such as employment status or husband’s social class, to describe the relationship between socioeconomic factors and hysterectomy more accurately.

## 5. Conclusions

There are social inequalities involved for women undergoing hysterectomy, as seen in this study, consistent with findings of studies in other countries. Hysterectomy is the most definitive surgical treatment method that can manage benign diseases, such as abnormal uterine bleeding and uterine fibroids. However, most women want to preserve their uterus, even after childbearing is complete. In fact, hysterectomy is not always good for the health of middle-aged women. The fact that the socially disadvantaged group has higher rates of hysterectomy suggests that the gynecologic screening strategy needs to be improved.

## Figures and Tables

**Figure 1 healthcare-10-00997-f001:**
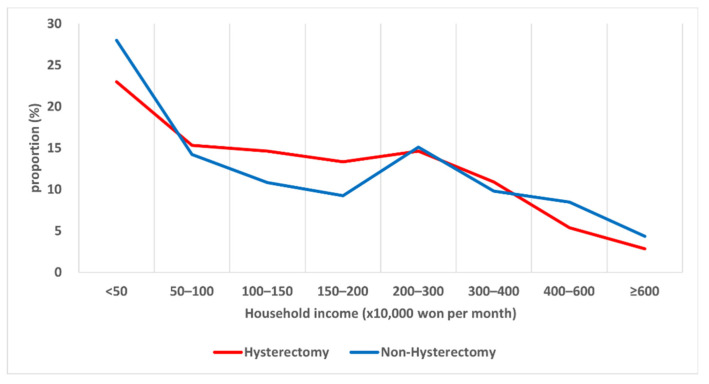
The proportion of participants with hysterectomy or without hysterectomy according to household income.

**Table 1 healthcare-10-00997-t001:** Comparison of characteristics between hysterectomy and non-hysterectomy groups.

Variables	Hysterectomy	Non-Hysterectomy	*p*-Value
No.	720	4552	
Age	54.63 (±8.90)	61.23 (±9.93)	<0.001
Education			0.164
Elementary school	313 (43.78)	2070 (45.98)	
Middle school	173 (24.20)	1037 (23.03)	
High school	193 (26.99)	1118 (24.83)	
College or higher	36 (5.04)	277 (6.14)	
Marital status			<0.001
No	9 (1.25)	22 (0.48)	
Married	603 (83.87)	3440 (75.80)	
Widow	90 (12.52)	956 (21.07)	
Separated	1 (0.14)	30 (0.66)	
Divorced	16 (2.23)	79 (1.74)	
unknown	0 (0.00)	11 (0.24)	
Household income (×10,000 KRW)			<0.001
<50	162 (22.98)	1221 (28.00)	
50–100	108 (15.32)	620 (14.22)	
100–150	103 (14.61)	472 (10.82)	
150–200	94 (13.33)	403 (9.24)	
200–300	103 (14.61)	658 (15.09)	
300–400	77 (10.92)	427 (9.79)	
400–600	38 (5.39)	370 (8.48)	
≥600	20 (2.84)	190 (4.36)	
Insurance			0.314
Unknown	7 (0.97)	38 (0.84)	
National insurance	658 (91.52)	4069 (90.00)	
Medicaid	54 (7.51)	414 (9.16)	
Parity	4.66 (±2.60)	4.77 (±2.42)	0.131
Age of menarche	15.85 (±1.89)	15.87 (±1.86)	0.781
Age of first pregnancy	23.66 (±3.31)	23.80 (±3.28)	0.264
Age of first delivery	30.49 (±4.29)	31.35 (±4.70)	<0.001
Breastfeeding			0.446
Never	48 (7.44)	258 (6.63)	
Done	597 (92.56)	3633 (93.37)	
BMI	25.09 (±3.25)	24.70 (±3.38)	0.001
Postmenopausal hormone replacement therapy	74 (10.57)	104 (2.31)	<0.001

Data are presented as mean ± standard deviation or N (%). BMI, body mass index; KRW, Korean won.

**Table 2 healthcare-10-00997-t002:** Univariate and adjusted multivariate logistic regression analyses of hysterectomy.

	Univariate			Multivariate *		
	Odds Ratio	95% Confidence Interval	*p*-Value	Odds Ratio	95% Confidence Interval	*p*-Value
Education level						
High school or lower	1.736	1.205–2.500	0.003	1.479	1.018–2.146	<0.05
College or higher	1.000			1.000		
Household income (×10,000 KRW)						
<400	2.268	1.704–3.021	<0.001	2.193	1.639–2.933	<0.001
≥400	1.000			1.000		
Insurance						
National insurance	1.155	0.853–1.565	0.352	1.314	0.958–1.805	0.090
Medicaid	1.000			1.000		

* Data are adjusted by age, body mass index, household income, insurance, and education level. KRW, Korean won.

## Data Availability

Raw data were generated from the Korea Genome and Epidemiology Study (KOGES). Derived data supporting findings of this study are available from the corresponding author upon reasonable request.

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
