# Peer review of "Analysis of the Relationship between Socioeconomic Status and Incidence of Hysterectomy Using Data of the Korean Genome and Epidemiology Study (KoGES)"

_healthcare, 2022, doi:10.3390/healthcare10060997_

Round 1

Reviewer 1 Report

General Impression:

The authors conducted a study to investigate the association between hysterectomy and the socioeconomic status of patients. The major strength of this study is that it included a large sample size. However, the authors did not adjust for the gynecologic/obstetric disorder or disease the patients had and did not mention how they allocated the control group of this study. In other words, it is not clear whether the control consisted of patients with or without gynecologic disorders. The use of the Chi-Square test to compare marital status, household income, and educational level is misleading. This test is best used for the comparison of 2x2 and 2x3 tables and to a lesser extent 3x3 tables. Nonetheless, the regression model is more reliable. Although the study did not investigate the differences in therapeutic approaches for the same disease in patients from different socioeconomic and educational backgrounds, it still gives a “bird's eye view” of the association between hysterectomy and these factors.

The abstract gives a clear and concise summary of the study. The introduction gives a good background about the topic and explains clearly the rationale of the study. The methods are described sufficiently but more details should be mentioned. The discussion is thorough and enjoyable. 

Comments:

1) In the introduction section (line 36), please correct the word “Dara” to “Data”.

2) In the introduction section (line 42), please correct the word “60 s” to “60s” (there is an extra space that should be removed).

3) In the introduction section (line 49), please specify whether hysterectomy is associated with higher or lower socioeconomic status and educational level according to the literature.

4) In the study population paragraph of the materials and methods section, please clarify whether this study retrospectively analyzed prospectively collected data or data were collected and analyzed prospectively. I.e., were the initial questionnaires that were designed in 2001 tailored for the purpose of this study or this is a posthoc analysis? I could not understand this point from the current context.

5) In the study population paragraph of the materials and methods section, please indicate whether or not the study included patients with obstetric complications (like placenta accrete spectrum, placenta previa, uterine atonia, placental abruption, etc…).

6) In the study population paragraph of the materials and methods section, please explain how you allocated the control group to this study. Did you match patients with the same obstetric/gynecologic disorders? If not, please explain this point in more detail.

7) In the data collection paragraph of the materials and methods section, please indicate whether or not you collected data about gynecological diseases (uterine leiomyoma, endometrial polyps, endometrial hyperplasia, etc…).

8) In the discussion section (line 133), please correct the word “out” to “our”.

9) In the discussion section (line 152), please correct the phrase “which in turn can to the development of habit for health promotion”.

10) The manuscript requires minor language revisions. Please revise carefully the written language.

Author Response

Thank you for your thoughtful review, and I have put the answer in the attached file.

Reviewer 2 Report

The study performed by Yung-Taek Ouh et al. gave an insight into the influence of socioeconomic on the incidence of hysterectomy in Korea. It drives the attention on the need of improving medical education and medical care accessibility to the low-income, low-education level population, which are regarded as a high-risk group of undergoing hysterectomy surgery in Korea by the conclusion of this study. Overall, it is a good study, but with some points need to be improved or well-explained.

  1. Study population: Which kind of chronic disease does the investigated population have? Why the sample size of the control group is so large comparing to the hysterectomy group, usually not exceeding 4 times. Considering the significant age difference, is it possible to balance the sample size by screening the control population by their age, to minimize the age difference between groups?
  2. The vertical legend ‘percentage of hysterectomy’ of Figure 1 is problematic. Non-Hysterectomy means no hysterectomy surgery, then it should always be 0.
  3. English writing needs to be improved: Especially in the Discussion section: Line 149, please re-write. Line 151-152, please re-write this sentence. Line 179, should be: have difficulty in visiting.
  4. In line 118, 119 and other related sentences throughout the text, to describe education level, ‘lower’ might be better than ‘lesser’, ‘higher’ might be better than ‘more’, please consider it.
  5. If how the cost of medical care (at least for the investigated population in the paper) in Korea is covered (self-paid, government-paid, partially self-paid and the percentage, partially government paid and the percentage, how insurance works, et al) could be introduced in the introduction, it will be better/stronger in supporting the conclusion by excluding the interference from this variance.
  6. The influence of conducting hysterectomy on woman’s life quality is inadequately described/emphasized, which reflects the meaning/significance of this study.
  7. The discussed points in the Discussion section are somehow not well organized, if it could be re-organized in a more logical way could be more interesting to read.
  8. Is there any extendable recommendation to other countries than Korea basing on the conclusion?

Author Response

(The authors gave the same response as above.)
